# Differential Alternative Splicing Genes and Isoform Regulation Networks of Rapeseed (*Brassica napus* L.) Infected with *Sclerotinia sclerotiorum*

**DOI:** 10.3390/genes11070784

**Published:** 2020-07-13

**Authors:** Jin-Qi Ma, Wen Xu, Fei Xu, Ai Lin, Wei Sun, Huan-Huan Jiang, Kun Lu, Jia-Na Li, Li-Juan Wei

**Affiliations:** 1College of Agronomy and Biotechnology, Southwest University, Chongqing 400715, China; jinqima1996@163.com (J.-Q.M.); 18722975519@163.com (W.X.); xf9310955@163.com (F.X.); ai_linnn@163.com (A.L.); reginasw@163.com (W.S.); jh13526462127@163.com (H.-H.J.); drlukun@swu.edu.cn (K.L.); 2Academy of Agricultural Sciences, Southwest University, Chongqing 400715, China

**Keywords:** *Brassica napus*, *Sclerotinia sclerotiorum*, alternative splicing, RNA-seq, WGCNA

## Abstract

Alternative splicing (AS) is a post-transcriptional level of gene expression regulation that increases transcriptome and proteome diversity. How the AS landscape of rapeseed (*Brassica napus* L.) changes in response to the fungal pathogen *Sclerotinia sclerotiorum* is unknown. Here, we analyzed 18 RNA-seq libraries of mock-inoculated and *S. sclerotiorum*-inoculated susceptible and tolerant *B. napus* plants. We found that infection increased AS, with intron retention being the main AS event. To determine the key genes functioning in the AS response, we performed a differential AS (DAS) analysis. We identified 79 DAS genes, including those encoding splicing factors, defense response proteins, crucial transcription factors and enzymes. We generated coexpression networks based on the splicing isoforms, rather than the genes, to explore the genes’ diverse functions. Using this weighted gene coexpression network analysis alongside a gene ontology enrichment analysis, we identified 11 modules putatively involved in the pathogen defense response. Within these regulatory modules, six DAS genes (ascorbate peroxidase 1, ser/arg-rich protein 34a, unknown function 1138, nitrilase 2, v-atpase f, and amino acid transporter 1) were considered to encode key isoforms involved in the defense response. This study provides insight into the post-transcriptional response of *B. napus* to *S. sclerotiorum* infection.

## 1. Introduction

Rapeseed (*Brassica napus* L.), the second largest oilseed crop in the world [1], is vulnerable to the necrotrophic ascomycete pathogen *Sclerotinia sclerotiorum*, which targets over 400 host species [2,3], including soybean (*Glycine max*), peanut (*Arachis hypogaea*), and sunflower (*Helianthus annuus*) [4]. *S. sclerotiorum* infects the leaves and stem of its host plants and causes Sclerotinia stem rot (SSR), also known as white mold, in rapeseed, resulting in yield losses of up to 94% in severe SSR outbreak seasons in the major growing areas such as Canada, China and the USA [5,6]. Despite this huge economic impact, no highly resistant *B. napus* varieties have been developed.

Current research into SSR is mainly focused on the pathogenicity and resistance mechanisms involved. The outcome of SSR is the product of the interaction of multiple factors in both *B. napus* and *S. sclerotiorum*, resulting in a gradient of resistance phenotypes termed quantitative disease resistance [7,8,9]. The resistance strategy has been frequently explored using approaches such as genome-wide association studies [10], quantitative trait locus mapping [11], RNA sequencing (RNA-seq) [12] and the identification of relevant noncoding RNAs [13], resulting in the discovery of rare but highly effective factors. Previous RNA-seq studies mainly focused on e transcriptional response and differential expression analyses. However, here, we use RNA-seq data to analyze alternative splicing (AS) to shed light on *B. napus* resistance to SSR.

AS is a type of post-transcriptional regulation that produces various mRNA transcripts from precursor mRNAs (pre-mRNAs) of a single gene, increasing transcriptomic and proteomic diversity [14]. AS occurs in more than 95% of intron-containing genes in humans [15] and ca. 33–60% of these genes in plants [16,17], with the types of AS including exon skipping (ES), intron retention (IR), alternative 5′ splice sites, alternative 3′ splice sites, mutually exclusive 3′ untranslated regions, tandem untranslated regions, mutually exclusive 5′ exons and mutually exclusive exons [18]. Although the ratios of these different AS types vary among different species, plant AS events predominantly involve IR, which is the least common type in animals [19].

AS occurs following the recognition by the spliceosome, containing small nuclear RNAs, of four cis-acting sequences: the exon–intron junctions at the 5′ and 3′ ends of the introns, the BRANCH-SITE sequence, and the polypyrimidine tract [20]. Strong recognition of certain splice sites usually results in constitutive splicing [21], which is generally based on the GU-AG rule [22]. The selection of splicing sites leading to AS is determined by RNA-binding proteins [23], mainly serine/arginine-rich (SR) proteins and heterogeneous nuclear ribonucleoprotein particle proteins, which recruit the spliceosome to bind to the splicing regulatory elements, such as the stimulating (exonic/intronic splicing enhancers) or inhibiting (exonic/intronic splicing silencers) splice sites [24].

AS affects the biology of a cell or organism by causing changes in mRNA stability, mRNA localization and mRNA translation, which further leads to noncoding transcripts or differently functional proteins and other biochemical properties [25]. AS may be associated with 15% of genetic diseases in humans [26]. However, it can also play important roles in plant development and stress response. In the area of pathogen defense, the AS of many *R* genes, including *TIR-NBS-LRR*, *Arabidopsis RPS4*, *Medicago truncatula RCT1,* Flax *L6*, Tomato *Bs4, CC-NBS-LRR* and so on, is required for the defense response [27] such as the production of the shorter and longer transcripts of the tobacco (*Nicotiana* sp.) *N* gene for resistance to the tobacco mosaic virus [28]. The diverse AS isoforms of *Arabidopsis thaliana* resistance to pseudomonas syringae 4 provide resistance to tomato strain DC3000 [29], whereas the diverse AS isoforms of *OsWRKY62* and *OsWRKY76* in rice (*Oryza sativa*) convey resistance against the pathogens *Magnaporthe oryzae* and *Xanthomonas oryzae* pv*. oryzae* [30]. In addition to identifying single AS events in a few genes in the past, recent studies have analyzed massive AS events and explored genome-wide AS networks; for example, massive and rapid AS responses were identified in the first few hours of cold stress in *A. thaliana* [14], while in the model grass *Brachypodium distachyon*, infection by the panicum mosaic virus and its satellite virus significantly increased the number of AS events [19]. The AS landscape in *B. napus* in response to *S. sclerotiorum* has not been reported.

In this study, we performed a differential AS analysis of *B. napus* RNA-seq data to explore candidate resistant genes in response to a *S. sclerotiorum* infection. We used coexpression analysis to identify the regulatory networks of the AS isoforms based on isoform expression levels rather than gene expression levels. This study elucidated the dynamic response of AS isoforms to enhance our understanding of the *B. napus* response to *S. sclerotiorum*.

## 2. Materials and Methods

### 2.1. Transcriptome Analysis

The RNA-seq data of *B. napus* before and after inoculation with *S. sclerotiorum* were downloaded from the GEO database (https://www.ncbi.nlm.nih.gov/geo/; accession numbers SRR3537150, SRR3537151, SRR3537153, SRR3537154, SRR3537155, SRR3537156, SRR3537159, SRR3537160, SRR1793826, SRR1793858, SRR1793863, SRR1793864, SRR1793865, SRR1793876, SRR1793909, SRR1793924, SRR1793925, and SRR1793927) [12,31]. The raw data were converted to the FASTQ format using the SRA Toolkit version 2.9.0 (https://www.ncbi.nlm.nih.gov/books/NBK158900/), and quality trimming and adapter clipping were then performed using Trimmomatic version 0.36 [32] (modified parameters: ILLUMINACLIP = TruSeq3-PE.fa:2:30:10 or ILLUMINACLIP = TruSeq3-SE.fa:2:30:10, LEADING = 3, TRAILING = 3, SLIDINGWINDOW = 4:15, and HEADCROP = 10). By generating the index using the chromosome sequence from *B. napus* Genome Browser (http://www.genoscope.cns.fr/brassicanapus/) [1], the prefiltered reads were mapped to the *B. napus* reference genome using STAR version 2.5.3a (with sjdbOverhang = 150) [33]. The mapped reads were assembled into putative transcripts using Cufflinks version 2.2.1 [34], then were tracked across all samples using Cuffcompare to estimate their relative abundances.

### 2.2. AS Landscape and Differential AS (DAS) Analysis

Using the gtf annotation obtained from Cufflinks, the AS landscape (intron retention, alternate acceptors, alternate donors, exon skipping, and other splicing events) of every sample was analyzed using Astalavista version 4.0 [35]. The DAS genes between different cultivars and different treatments were respectively quantified using rMATS (with cstat = 0.0001, tstat = 6, FDR < 0.01) [36], and were then visualized using rmats2sashimiplot-master (https://github.com/Xinglab/rmats2sashimiplot).

### 2.3. Weighted Gene CoExpression Network Analysis (WGCNA) of Isoforms

To further explore the regulatory relationship of each isoform, a WGCNA [37,38] run in R version 3.4.4 was used to perform data cleaning (the top 14% transcripts were selected based on their fragments per kilobase of transcript per million mapped reads (FPKM) variance analysis), sample clustering, soft threshold filtering (power = sft$powerEstimate), scale-free check (scaleFreePlot), one-step network construction (TOMType = “unsigned”, mergeCutHeight = 0.25, verbose = 3), module identification, and network visualizing (random selection of 1000 transcripts). The weighted values of each relationship between two coexpressed transcripts gained from each module were filtered with > 0.15 to obtain the significantly coexpressed networks using Cytoscape version 3.5.1 [39]. Using R, the isoform heatmap was constructed on the average FPKM values of the biological replicates, which were transformed into log_2_ (FPKM + 1) values.

### 2.4. Functional Enrichment and Clustering

To determine the potential function of each module, a gene ontology (GO) enrichment analysis was performed using the R package topGO [40]. The modules containing fungus relative terms with classicFisher < 0.01 were selected as target modules, such as response to fungus and defense to fungus. A GO enrichment Senior Bubble plot was drawn by the OmicShare tools (http://www.omicshare.com/tools).

### 2.5. Homologous Gene Identification

Based on the studies showing that *A. thaliana* and *B. napus* have a common ancestor in the Brassicaceae [41,42], the arabidopsis genes homologous to the *BnDAS* genes were identified using BLASTP with an *E* value of ≤ e^−5^ using the NCBI BLAST version 2.2.30+ program [43].

## 3. Results

### 3.1. AS Landscape in B. napus in Response to S. sclerotiorum

To identify the AS landscape of *B. napus* in response to a *S. sclerotiorum* infection, a total of 18 RNA-seq libraries were analyzed in this study (Table 1). The raw data were downloaded from two datasets, one of which comprised single-end reads and the other paired-end reads. The first dataset contained eight RNA-seq libraries from the leaves of two *B. napus* cultivars (susceptible Westar and resistant ZY821) at 24 h postinoculation (hpi) with *S. sclerotiorum* or a mock treatment. The second dataset contained ten libraries from the stems of two *B. napus* cultivars (susceptible J902 and resistant J964) subjected to a mock treatment or at 24 and 48 hpi with *S. sclerotiorum*. Based on the reference genome [1], we performed a quality trim, genome mapping, transcript assembly and transcript merging to identify the FPKM of every isoform (Appendix A). For the single-end transcriptome data, 70.31% of the reads were uniquely mapped to the reference genome and the average mapped read length was 96.32 bp. For the paired-end transcriptome data, 82.55% of the reads were uniquely mapped to the reference genome and the average mapped read length was 178.26 bp (Appendix A).

The assembled transcript annotation file was further used to identify the AS landscape using Astalavista. As shown in Figure 1 and Appendix A, the results indicated that IR was the most common AS event compared with the AA (alternate acceptors), AD (alternate donor), and ES. In addition, we computed the AS event frequency of the merged transcripts (overall AS events/merged transcript number). Compared with the mock treatment (Figure 1a,c,e,g), the AS ratio was slightly increased in the 24 hpi plants (Figure 1b,d,f,h), whereas the AS ratio was decreased in the 48 hpi treatment (Figure 1i). This indicated that *S. sclerotiorum* triggered AS at the early stages of infection.

### 3.2. Identification of DAS Genes

In total, 160 DAS genes were identified from the paired-end RNA libraries using rMATS, whereas no DAS genes were identified from the single-end data. We filtered the 160 DAS genes using a false discovery rate (FDR) < 0.05, resulting in the identification of 79 important DAS genes. Using a differential analysis of the mock-treated and infected materials, the number of infection-induced DAS genes in the comparisons J2_24 h vs. J2_ck, J4_24 h vs. J4_ck, and J4_48 h vs. J4_ck were, respectively, 22, 6, and 10 (Appendix A). Differential analysis between the resistant and susceptible cultivars (J2_24 h vs. J4_24 h) after inoculation with *S. sclerotiorum* led to the identification of 49 cultivar-related DAS genes. Eight overlapping DAS genes were detected in both infection-induced and cultivar-related DAS genes.

Homologs of 68 of the *B. napus* DAS genes were identified in arabidopsis (Table 2), which enabled their annotation; the functions of the other 11 *B. napus* DAS genes remained unknown. In addition, 27 of the arabidopsis homologous genes were previously shown to be AS genes, according to the ASIP database [44]. Based on the GO enrichment analysis, the 79 significantly DAS genes were involved in a variety of functions including the valine catabolic process, nuclear retention of unspliced pre-mRNA, nuclear mRNA surveillance of the spliceosome and fatty acid β-oxidation (Appendix A).

Based on the annotated functions of the homologous genes in *A. thaliana*, four of the 79 DAS genes were putatively related to RNA splicing, including *BnaCnn1978* (RNA-binding protein), *BnaC08g21130D* (SR protein 34a), and *BnaC07g01360D* (RNA-binding protein). One gene (*BnaA09g50970D*, splicing factor PRP18 protein), which was previously shown to be involved in promoting splicing at weak splice sites [45], was identified in both the infection-induced and cultivar-related DAS gene groups. Three of the 79 DAS genes were known to be related to the defense response, including *BnaA05g20710D* (immune-associated nucleotide binding 9), which encodes a plasma membrane-localized protein that functions as a negative regulator of basal immunity in *A. thaliana* [46]; *BnaC09g07670D* (LRR [Leucine-rich repeat] protein), which is reported likely as a pseudogene with its two paralogs transducing the defense signal downstream of TNL(TIR -type sensor NLR) in *A. thaliana* [47]; and *BnaC04g37090D* (immunoglobulin E-set superfamily protein). Two genes, *BnaA07g13990D* (encoding a protein related to AP2.7) and *BnaA03g06980D* (transcription regulator NOT2/NOT3/NOT5 protein), were believed to be involved in transcriptional regulation. The cultivar-related AS genes were *BnaC05g22390D* (RNA polymerase II degradation factor-like protein) and *BnaA05n0101* (FRS [FAR1-RELATED SEQUENCE] family protein). In addition, we identified DAS genes encoding some important enzymes, such as protein kinases, nitrilase and transferases.

### 3.3. WGCNA of All Isoforms

WGCNA is used to describe gene correlation patterns and identify key genes by detecting gene modules, intramodular connectivity, gene significance and other features [38]. To explore the potential function of the DAS genes and analyze the diverse functions of the different isoforms encoded by a single gene, we performed a WGCNA using 575,171 isoforms (Figure 2). The FPKM values of all transcripts, including all isoforms, were next used to generate the coexpression network.

We performed a variance analysis to remain 40,441 isoforms so that the soft threshold could be up to five. Based on the distribution that approximately followed a straight line (Figure 2c), the network was a scale-free topology, meaning it was suitable for use in the coexpression analysis. We identified 85 modules (Appendix A), then randomly selected 1000 isoforms to plot the network heatmap.

We performed a GO analysis on each of the identified modules (Appendix A), and 11 modules were selected for a further analysis to identify the response-related isoforms based on the GO terms and defense response (Figure 3). Among the 11 pathogen-related modules, we identified 25 response-related isoforms whose precursor genes were identified as 19 DAS genes (Figure 4, Appendix A). Three of these 19 genes (BnaC05g05550D, BnaC08g21130D, and BnaC05g00740D) encoded nine isoforms that were identified in several modules, which suggested that these genes might play crucial roles in different response-related networks by producing different isoforms. In addition, another three overlapping DAS genes (BnaA06g38980D, BnaC02g27060D, and BnaA03g58530D) were identified as being differentially spliced in both infection- and cultivar-specific manners. These six DAS genes were considered to be the key candidate defense genes on which to perform further analysis.

### 3.4. The Isoform Expression Landscape of Crucial DAS Genes

To further elucidate the function of the above six candidate genes, we illustrated their DAS pattern in terms of their genomic coordinates and the expression pattern of their isoforms. In Figure 5, the reads of the ES isoforms and exon inclusion isoforms were quantified for each candidate gene to determine inclusion levels (IncLevels), representing the normalized proportion of exon inclusion events. In addition to illustrating the proportion of AS events, Figure 5 also provides modified values for the number of reads per kilobase of transcript per million mapped reads (RPKM) (y axis) of every site along the genomic coordinate (x axis) and the junction reads (the number labeled on the line) to present the read density [48].

*BnaC05g05550D* is homologous to *AT1G07890*, which encodes ascorbate peroxidase 1 (APX1). This gene was DAS in the susceptible and resistant cultivars at 24 hpi with *S. sclerotiorum* (Figure 5a). In susceptible cultivar J2, the IncLevel of *BnaC05g05550D* was much lower than it was in the resistant cultivar J4, whereas the first and the third exons were expressed to a much greater level. This suggests that, in the resistant cultivar, APX1 tended to reduce the ES events so that as many exons as possible were retained in the transcripts, whereas the susceptible cultivar tended to skip its second exon. Expression heatmaps of all 13 isoforms of APX1 were made (Figure 6a), which were differently abundant in the different conditions analyzed. Only five isoforms (TCONS_00410992, TCONS_00415336, TCONS_00420888, TCONS_00420889, and TCONS_00420890) were involved in the defense-related modules (turquoise, yellow-green, and cyan modules). This suggested that these five defense-related isoforms may function differently in the two cultivars when infected with *S. sclerotiorum*, resulting in differences in their resistance levels to this pathogen.

*BnaC08g21130D* is homologous to *AT3G49430*, which encodes SR34a. This gene was DAS in the mock-treated and infected resistant cultivar plants at 48 hpi (Figure 5b). In the infected material, the IncLevel was much lower than in the mock-treated material, and the first and third exons were expressed to a much higher level. This suggests that, in response to the infection, the ES events in *SR34a* were increased, with the second exon being skipped. Expression heatmaps were made of all 19 isoforms of *SR34a* (Figure 6b). Only five isoforms were identified in the two different materials above, with the TCONS_00494944 isoform involved in a defense-related network (turquoise). This suggested that the five isoforms might respond to the infection in the resistant cultivar after 48 h.

*BnaC05g00740D* is homologous to *AT1G01170*, which encodes the protein of unknown function 1138. The *BnaC05g00740D* transcripts were differentially alternatively spliced in the mock-treated and infected susceptible cultivar at 24 hpi (Figure 5c). Under the infection, the IncLevel was much higher than in the mock-treated plants, suggesting that *BnaC05g00740D* reduces the ES events to retain all the exons at 24 hpi. Expression heatmaps were made of all eight isoforms of *BnaC05g00740D* (Figure 6c). Four of the isoforms showed different abundances in the different experimental conditions. The isoforms associated with the defense-related network were not significantly differently abundant between the different conditions, however. This suggested that the four differently abundant isoforms may have novel roles in responding to *S. sclerotiorum* infection.

*BnaA06g38980D* is homologous to *AT3G44300*, which encodes nitrilase 2 (NIT2). This gene was DAS in the susceptible and resistant *B. napus* cultivars at 24 hpi with *S. sclerotiorum*, and between the mock-treated and 24 hpi susceptible cultivar plants (Figure 5d). In the infected susceptible cultivar J2, the IncLevel was much lower than in the infected resistant cultivar J4, suggesting that the resistant cultivar tended to reduce the number of ES events to retain as many exons as possible. The susceptible cultivar J2 had a much higher IncLevel when infected with *S. sclerotiorum* for 24 h than it did in the mock-treated condition, suggesting that it tends to reduce the ES events to retain all exons during infection. Expression heatmaps were made of all 16 isoforms of *BnaA06g38980D* (Figure 6d), enabling the identification of five isoforms that significantly differed in abundance between the different experimental conditions. After infection, TCONS_00150596 and TCONS_00150597 were detected in the susceptible cultivar, whereas TCONS_00151240 from the turquoise module only occurred in the resistant cultivar, suggesting that the isoforms that are specific to the cultivars may be key factors in the defense response.

*BnaC02g27060D* is homologous to *AT4G02620*, which encodes the vacuolar ATPase subunit F family protein (V-ATPase F). This gene was DAS in the mock-treated resistant cultivar and the same plants at 48 hpi with *S. sclerotiorum*, and between the two different cultivars at 24 hpi (Figure 5e). In the susceptible cultivar J2 at 24 hpi, the IncLevel was much higher than it was in the infected resistant cultivar J4, suggesting that the retention of the second exon in the resistant cultivar resulted in its high level of resistance to this pathogen. When compared with the mock-treated plants, the IncLevel of V-ATPase F was much lower in the resistant cultivar at 48 hpi, suggesting that infection may stimulate the ES events that generate the diverse isoforms. Expression heatmaps were made of all 13 isoforms of *BnaC02g27060D* (Figure 6e). Six isoforms were significantly differently abundant in the different materials. TCONS_00301858 was included in the darkseagreen4 module expressed in all the susceptible cultivars, but not in the resistant cultivars, which may suggest that this isoform was specific to the susceptible cultivar. None of the isoforms were expressed in the mock-treated resistant cultivars, indicating that V-ATPase F was only expressed in the resistant cultivars when induced by the pathogen.

*BnaA03g58530D* is homologous to *AT4G21120*, which encodes an amino acid transporter 1 (AAT1). This gene was DAS in the mock-treated and infected resistant cultivar J4 at 24 hpi, and between the susceptible and resistant cultivars at 24 hpi (Figure 5f). Compared with the infected resistant cultivar, the infected susceptible cultivar and the mock-treated resistant cultivar both had higher IncLevels. More ES events may therefore occur in the infected resistant cultivar, which may therefore play a greater role in the infection response. Expression heatmaps were made of all four isoforms of *BnaA03g58530D* (Figure 6f). Two of these isoforms were significantly differently abundant in the different materials. TCONS_00080439 from the turquoise module occurred in both of the cultivar-related and infection-induced DAS groups, suggesting that it might be a key isoform responsible for enhancing resistance.

## 4. Discussion and Conclusions

In this study, we aimed to identify genetic factors and specific response mechanisms that influence resistance to SSR by analyzing the RNA-seq data of infected and mock-treated resistant and susceptible cultivars. Two types of analysis were performed: post-transcriptional differences were explored using a DAS analysis, and isoform-level transcriptional regulation was elucidated using a coexpression network analysis. A total of 79 DAS genes were identified, which was fewer than expected. However, a previous study showed that the use of ten differential splicing analysis tools resulted in the identification of different numbers of DAS genes. The number of identified DAS genes ranged from 0 (cuffdiff2) to 4506 (edgeR) in a human prostate cancer dataset, with 2962 and 0 DAS genes identified using rMATS in human and mouse datasets, respectively [49]. In addition, 252, 171, and 42 DAS genes were identified in comparisons between control and intolerant, control and tolerant and intolerant and tolerant catfish following heat stress [50]. Additionally, rMATS was used to identify 24, 40, and 74 DAS genes in the root, juvenile leaves and old leaves of *B. napus*, respectively, in response to boron deficiency [51].

The 79 DAS genes identified here were mainly involved in the nuclear retention of unspliced premRNA and the nuclear mRNA surveillance of the spliceosome. Using the same double-end RNA-seq data here, it has been reported that 13,276 differentially expressed genes (DEGs) participated in metabolic processes and responses to stimuli such as response to chitin, RNA methylation, secondary cell wall biogenesis and gravitropism [31]. However, these genes had little overlap with the 79 DAS genes identified here. A small overlap was identified between the DEGs and DAS genes in rice plants exposed to mineral (phosphorus, zinc, manganese, copper, and iron) deficiencies [52]. These comparisons suggested that AS is an additional layer of regulation, complementary to DEGs, that enables plants to respond to their environment.

*S. sclerotiorum* infects *B. napus* using mycelia, or through the production of apothecia which are laden with fungal spores [53]. The fungus produces multiple pathogenicity factors that attack *B. napus*, such as oxalic acid and an array of lytic enzymes [54]. When attacked by fungi, plants undergo a three-step process: pathogen perception, signal transduction and the defense response. In the plant–*Sclerotinia* system, pathogen detection is based on the recognition of pathogen-associated molecular patterns (PAMPs) by pattern recognition receptors, such as RLP30 (receptor-like protein 30) [7] and other receptor-like kinases, or via cytoplasmic leucine rich proteins (LRRs) [55]. The subsequent signal transduction involves a MAPK (mitogen-activated protein kinase) cascade or MAPK-independent pathways [56] involving many signaling molecules, including reactive oxygen species (ROS) [57], nitric oxide [58], salicylic acid [59], jasmonic acid and ethylene [60]. These signals drive plant PAMP-triggered immunity through the production of nuclear proteins including transcription factors and protein kinases that activate ROS production, detoxification, oxidative protection, callose deposition, camalexin production and the production of other specialized metabolites [61,62]. The DAS genes identified here, such as the immune-associated genes and those encoding important enzymes, might play a major role in the response to *S. sclerotiorum*.

Among the 79 DAS genes, we identified several genes, including SR, that are involved in RNA splicing. As constitutive and AS regulators, SR proteins consist of RNA recognition motif and a Ser/Arg-rich domain, which, respectively, recognize RNA and the spliceosome [63]. It has been reported that genes encoding SR proteins in plants appear to function mainly in stress responses; for example, in arabidopsis, the *sr34b* T-DNA insertion mutant had a lower cadmium tolerance with a splicing defect in its iron-regulated transporter 1 [64], whereas the *sr45* mutant was hypersensitive to glucose and abscisic acid [65]. In rice, the SR40, SCL57, and SCL25 proteins were important for phosphorus uptake and remobilization [52]. Recent studies have reported the stress-triggering AS pattern of certain *SR* transcripts, which may contain premature termination codons, resulting in their degradation by the nonsense-mediated decay pathway. For example, the abundance of an unproductive premature termination codon-containing AS isoform of arabidopsis SR30 decreases at elevated temperatures [66]. The AS of SR genes can increase plant tolerance to hormones and stress treatments in rice and *B. napus* [51,67,68]. It was interesting to discover that the gene encoding the splicing factor PRP18 protein (*BnaA09g50970D*) is an overlapping DAS gene, as the homologous mutant in *A. thaliana* produces short roots and small siliques following its induction of large genome-wide IR events [45]. Additionally, *BnaCnn1978*, encoding an RNA binding protein, was identified as an infection-induced DAS gene. The splicing-related gene *SR34a* was also involved in two defense-related modules, and the increased number of ES events at 48 hpi with *S. sclerotiorum* suggests that this crucial splicing factor might be activated in response to pathogen stress to regulate downstream networks. The regulation relationships should be verified in further experiments.

In addition to SR34a, we also identified other five DAS genes (APX1, NIT2, AAT1, V-ATPase F and a gene encoding an unknown protein) as crucial candidate genes in the response to pathogen defense. APX isoenzymes function in controlling ROS metabolism and eliminating H_2_O_2_ [69]. APX can respond to environmental stresses such as oxidative stresses [70,71], light stress [69] and pathogen infection (*Trichoderma harzianum*) [72,73]. In this study, we identified APX1 as a cultivar-related DAS with isoforms involved in different coexpression networks in the response to pathogen stress. NIT2 in *A. thaliana* is able to hydrolyze a broad substrate range [74], whereas a recent study revealed that AtNIT2 expression was de novo-induced by a plant pathogen (*Pseudomonas syringae* pv. tomato) and responsible for R gene-mediated resistance responses [75]. We established that *BnNIT2* was differently spliced in the susceptible cultivar at 24 hpi compared with the mock-treated plants, and seemed to reduce the ES events in the susceptible cultivar in response to *S. sclerotiorum*. V-ATPase F proton pumps are responsible for the ATP hydrolysis that drives ion transport [76]. It has been reported that V-ATPase functions in response to phosphorus deficiency [77], copper stress [76], fungal infection (the dermatophyte *Trichophyton rubrum*) [78] among other stresses. We identified V-ATPase F as a pathogen-induced and cultivar-related DAS gene, and its AS frequency increased at 48 hpi. AATs are responsible for the long-distance transport of amino acids from the root to the shoot [79]. The overexpression of AtAAT1 resulted in improved resistance against the pathogen *Pseudomonas syringae* [80]. In this study, we identified AAT1 as both a pathogen-induced and cultivar-related DAS gene, which might lead to the generation of more AS events at 24 hpi with *S. sclerotiorum* to increase plant resistance to this pathogen.

In this study, a total of 79 DAS genes were identified in *B. napus* after inoculation with *S. sclerotiorum*, and a combination of a DAS analysis and an isoform WGCNA revealed six candidate genes with important putative functions in pathogen resistance. These observations enhance our understanding of the post-transcriptional response of *B. napus* to *S. sclerotiorum* infection and pave the way to developing rapeseed lines with improved resistance to this devastating pathogen. Continuous advances in molecular biological experiments in exploring the function of different isoforms of one single gene should facilitate our findings of the candidate isoforms in pathogen defense.

## Figures and Tables

**Figure 1 genes-11-00784-f001:**
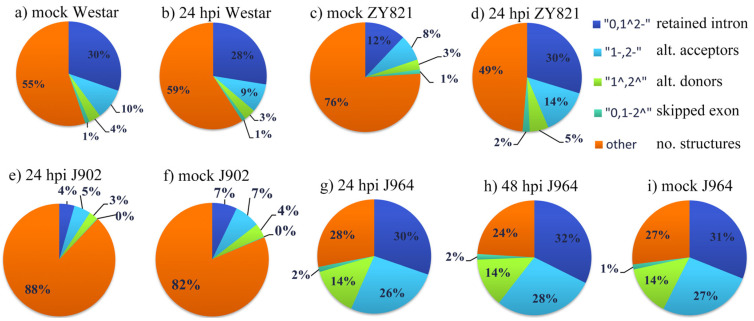
AS Landscapes of healthy and pathogen-inoculated *B. napus*. Frequency of AS types of mock (**a**,**c**) and 24 hpi (**b**,**d**) in *B. napus* susceptible (cv. Westar) and tolerant (cv. ZY821) genotypes. Frequency of AS types of mock (**f**), 24 hpi (**e**) in *B. napus* susceptible (cv. J902) and mock (**i**), 24 hpi (**g**), 48 hpi (**h**) in *B. napus* tolerant (cv. J964) genotypes.

**Figure 2 genes-11-00784-f002:**
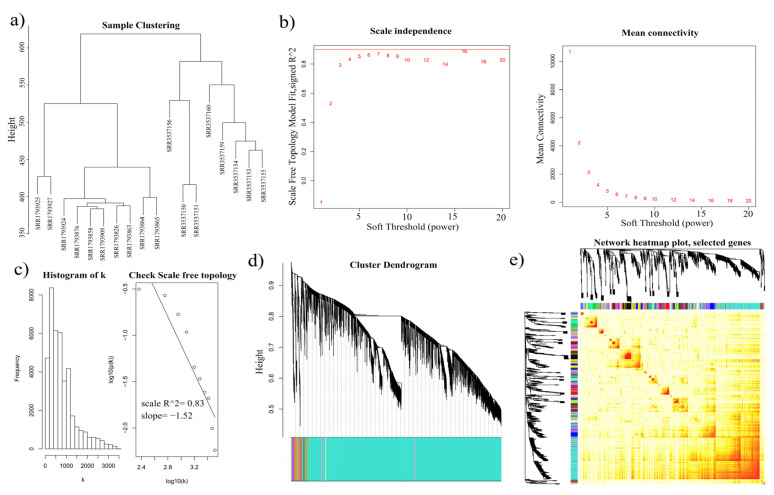
Coexpression network generation in isoform level by WGCNA. (**a**) Sample clustering with the FPKM of all isoforms from eighteen RNA-seq libraries. (**b**) Soft threshold selection based on the recommended power. (**c**) Scale free topology check with coefficient of determination (R^2^) equal to 0.83. (**d**) Cluster dendrogram of all isoforms by dissimilarity to obtain 85 modules. Each vertical line represents a single isoform. (**e**) Network heatmap of 1000 randomly selected isoforms.

**Figure 3 genes-11-00784-f003:**
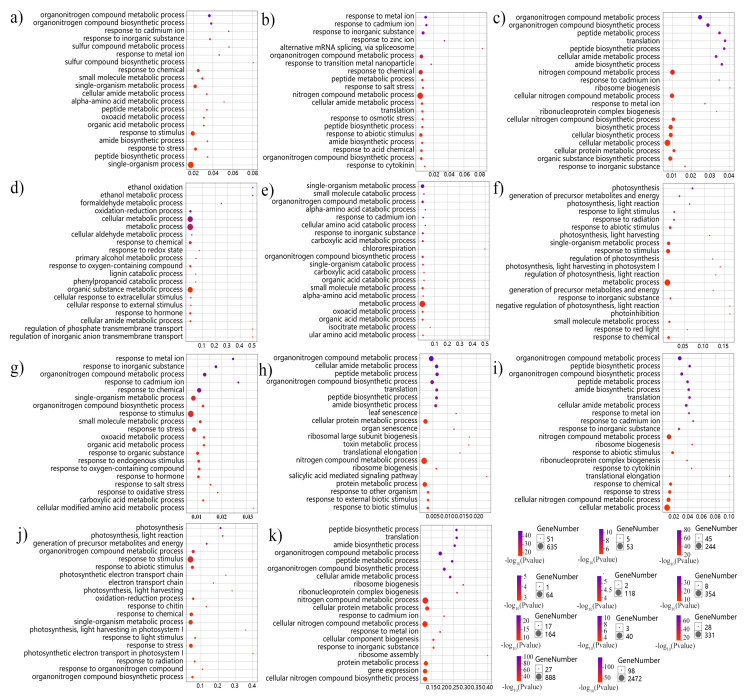
Top 20 of GO terms enrichment Senior Bubble of the eleven defense-related modules. (**a**–**k**) represents black, plum2, royalblue, coral1, yellowgreen, tan, lightyellow, darkseagreen4, cyan, blue, turquoise module. The *x*-axis represents the RichFactor which shows the enrichment level. The size of the bubble represents the enriched gene number.

**Figure 4 genes-11-00784-f004:**
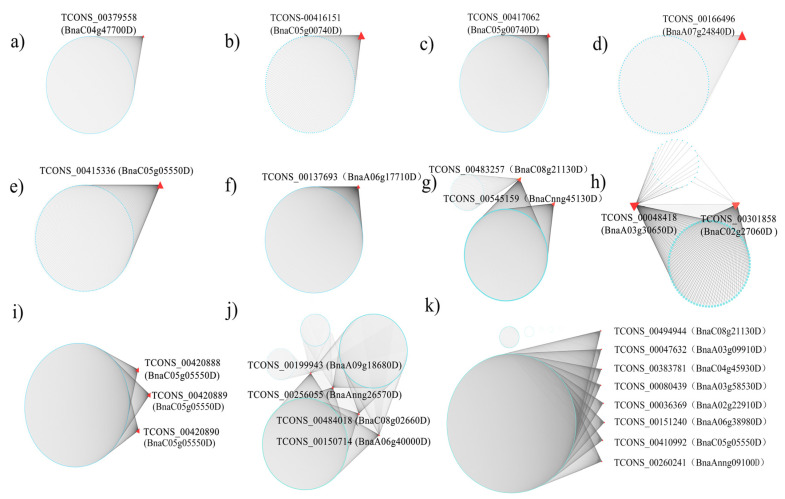
Coexpression networks of DAS genes in eleven defense-related modules. (**a**–**k**) represents black, plum2, royalblue, coral1, yellowgreen, tan, lightyellow, darkseagreen4, cyan, blue, turquoise module. Using Cytoscape, the red triangles represent the involved isoforms of the DAS genes, with the blue nodes representing connected isoforms.

**Figure 5 genes-11-00784-f005:**
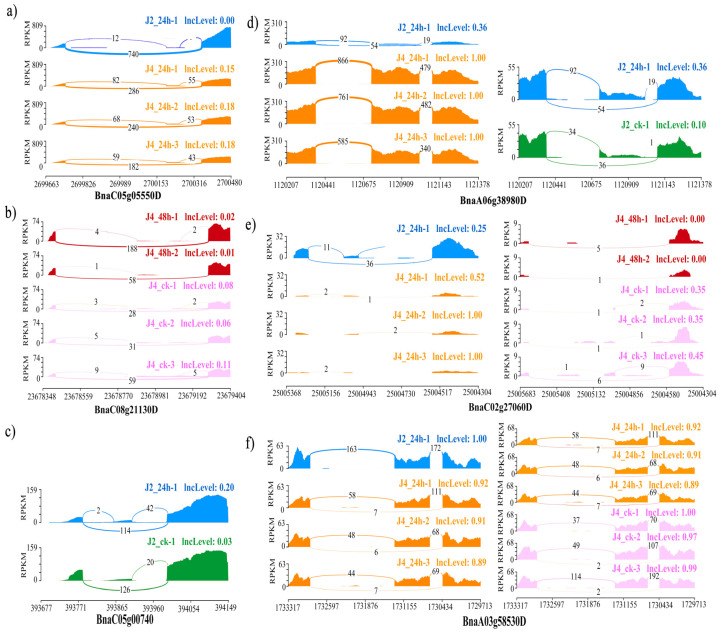
Quantitative visualization (sashimiplot) of the six candidate DAS genes. (**a**–**c**) illustrates the DAS between two groups of *BnaC05g05550D*, *BnaC08g21130D* and *BnaC05g00740D.* (**d**–**f**) illustrates the DAS between four groups of *BnaA06g38980D, BnaC02g27060D* and *BnaA03g58530D.* The different colors represent five different conditions: blue, cv. J902 at 24 hpi; red, cv. J964 at 48 hpi; pink, cv. J964 mock inoculation (ck); green, cv. J902 mock inoculation; and orange, cv. J964 at 24 hpi. The y axis represents a modified RPKM value. The x axis represents the genomic coordinates of each DAS gene. The numbers on the curved lines indicate the counts of each splice junction. The IncLevel value represents the normalized proportion of EI (Exon Inclusion) events.

**Figure 6 genes-11-00784-f006:**
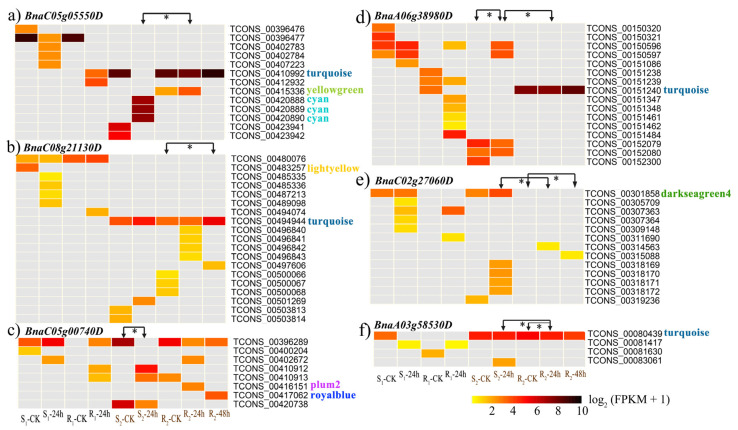
Expression heatmaps of each isoform of the six candidate DAS genes. (**a**–**k**) represents the isoform expression pattern of *BnaC05g05550D*, *BnaC08g21130D*, *BnaC05g00740D, BnaA06g38980D, BnaC02g27060D* and *BnaA03g58530D*. The arrows indicate the materials in which the DAS genes are significantly differently abundant. The colors written behind the isoforms represent their involved coexpression modules. S_1_-CK, mock-inoculated Westar plants; S_1_–24 h, Westar plants at 24 hpi; R_1_-CK, mock-inoculated ZY821 plants; R_1_–24 h, ZY821 plants at 24 hpi; S_2_-CK, mock-inoculated J902 plants; S_2_–24 h, J902 plants at 24 hpi; R_2_-CK, mock-inoculated J964 plants; R_2_–24 h, J964 plants at 24 hpi; R_2–_48 h, J964 plants at 48 hpi.

**Table 1 genes-11-00784-t001:** Statistical summary of RNA-seq material and the detected alternative splicing (AS) pattern.

Cultivar	Treatment	Dataset Name	Data Type	Number	Overall AS Events	Merged Transcript-Number	AS Event Ratio	AS Type				
								IR	AA	AD	ES	Other Event
Westar	Mock	W_ck	SE	SRR3537150, SRR3537151	28,518	585,673	4.87%	8638	2724	1049	380	15,727
Westar	24 hpi	W_24 h	SE	SRR3537153, SRR3537154, SRR3537155	44,388	784,570	5.66%	12,230	3860	1538	518	26,242
ZY 821	Mock	Z_ck	SE	SRR3537156	7997	234,401	3.41%	964	617	237	100	6079
ZY 821	24 hpi	Z_24 h	SE	SRR3537159, SRR3537160	16,770	467,414	3.59%	4993	2326	934	358	8159
J902	Mock	J2_ck	PE	SRR1793858	34,734	388,008	8.95%	2470	2538	1371	79	28,276
J902	24 hpi	J2_24 h	PE	SRR1793826	69,013	536,471	12.86%	3038	3311	1792	130	60,742
J964	Mock	J4_ck	PE	SRR1793924, SRR1793925, SRR1793927	32,750	622,164	5.26%	10,156	8688	4606	453	8847
J964	24 hpi	J4_24 h	PE	SRR1793863, SRR1793864, SRR1793865	34,397	639,525	5.38%	10,405	9084	4759	489	9660
J964	48 hpi	J4_48 h	PE	SRR1793876, SRR1793909	15,202	387,918	3.92%	4912	4308	2087	239	3656

**Table 2 genes-11-00784-t002:** Annotation of the 79 DAS genes in *B. napus.*

DAS Comparison	DAS Gene	At Ortholog	ASIP Synonym	% Identity	At Ortholog Family
J2_24 h vs. J2_ck	BnaC05g00740D	AT1G01170	T25K16.16	92.50	Protein of unknown function (DUF1138)
BnaA05g20710D	AT1G33970	F12G12.21	82.22	Immune-associated nucleotide binding 9
BnaA07g03760D	AT1G64610		57.47	Transducin/WD40 repeat-like superfamily protein
BnaA07g25520D	AT1G65950	F12P19.11	87.07	Protein kinase superfamily protein
BnaA09g18680D	AT2G01540		94.67	Calcium-dependent lipid-binding protein
BnaA02g26350D	AT2G02360		61.11	Phloem protein 2-B10
BnaA07g13990D	AT2G28550	T17D12.11	80.88	(TF) Related to AP2.7
BnaC04g04760D	AT2G46090	T3F17.26	84.18	Diacylglycerol kinase protein
BnaC03g25720D	AT2G46340		77.77	SPA (suppressor of phyA-105) protein
BnaC01g36600D	AT3G14770	T21E2.6	83.47	Vacuolar sugar transporter SWEET2
BnaCnng45490D	AT4G16845	FCAALL.23	80.65	VEFS-Box of polycomb protein
BnaCnn1978	AT4G35785	F4B14.7	68.05	RNA-binding (RRM/RBD/RNP motifs) protein
BnaA03g06980D	AT5G18230		88.12	Transcription regulator NOT2/NOT3/NOT5 protein
BnaA10g14420D	AT5G21170	T10F18.200	80.12	5’-AMP-activated protein kinase β-2 subunit protein
BnaA03g09910D	AT5G58110		88.38	ATPase activators
BnaA06g22260D	AT5G62760		71.72	P-loop containing nucleoside triphosphate hydrolases superfamily
BnaC09g07670D	AT5G66890		73.10	Leucine-rich repeat (LRR) protein
BnaC09g23900D	unknown			
J2_24 h vs. J2_ck\ J2_24 h vs. J4_24 h	BnaA09g50970D	AT1G03140		79.02	Splicing factor Prp18 protein
BnaC09g09280D	AT1G75660	F10A5.15	68.57	5′-3′ exoribonuclease 3
BnaA06g38980D	AT3G44300		81.42	Nitrilase 2
BnaC04g40450D	AT5G65940	K14B20.11	80.00	β-hydroxyisobutyryl-CoA hydrolase 1
J2_24 h vs. J4_24 h	BnaCnng45130D	AT1G02305		78.82	Cysteine proteinases superfamily protein
BnaC05g05550D	AT1G07890	F24B9.2	92.00	Ascorbate peroxidase 1
BnaC05g22390D	AT1G29350		74.24	RNA polymerase II degradation factor-like protein (DUF1296)
BnaA06n0344	AT1G48840		50.79	Plant protein of unknown function (DUF639)
BnaC06g04140D	AT1G50910		79.28	Hypothetical protein
BnaA02g12910D	AT1G67170		69.08	Sarcolemmal membrane-associated protein
BnaA07g24840D	AT1G68010	T23K23.14	96.90	Hydroxypyruvate reductase
BnaCnng00710D	AT2G04560		82.68	Transferases
BnaCnng63360D	AT2G19480	F3P11.8	84.7	Nucleosome assembly protein 1;2
BnaC04g33610D	AT2G21270		80.73	Ubiquitin fusion degradation 1
BnaAnng26570D	AT2G23010		77.02	Serine carboxypeptidase-like 9
BnaC04g45930D	AT2G26900		89.86	Sodium Bile acid symporter family
BnaC04g47700D	AT2G41460		69.98	Apurinic endonuclease-redox protein
BnaC04g03300D	AT2G44065		90.23	Ribosomal protein L2 family
BnaA03g30650D	AT3G09500		94.31	Ribosomal L29 protein
BnaAnng09240D	AT3G12130		78.88	KH domain-containing protein/zinc finger (CCCH type) family
BnaC09g01340D	AT3G27100	MOJ10.18	87.50	ENY2
BnaC02g09410D	AT3G44280		73.15	Peptidyl-prolyl cis-trans isomerase G
BnaA06g17710D	AT3G46970		95.01	α-glucan phosphorylase 2
BnaA04g04080D	AT3G54480	T14E10.50	85.87	SKP1/ASK-interacting protein 5
BnaC09g00230D	AT4G02780		85.71	Terpenoid cyclases/Protein prenyltransferases superfamily
BnaA05n0101	AT4G15090		84.82	(TF) FRS (FAR1 Related Sequences) family
BnaA01g17760D	AT4G17300		90.49	Class II aminoacyl-tRNA and biotin synthetases superfamily
BnaC03g02590D	AT5G06440	MHF15.4	61.86	Polyketide cyclase/dehydrase/lipid transport superfamily protein
BnaA06g26720D	AT5G24150	MLE8.1	77.76	FAD/NAD(P)-binding oxidoreductase protein
BnaA07g15350D	AT5G41350		79.44	RING/U-box superfamily protein
BnaA02g22910D	AT5G42210		81.51	Major facilitator superfamily protein
BnaA02g25070D	AT5G47180	MQL5.3	94.09	Plant VAMP (vesicle-associated membrane protein) family
BnaC09g28380D	AT5G52380		69.28	Vascular-related nac-domain 6
BnaC03g14120D	AT5G55700		89.87	β-amylase 4
BnaC09g33270D	AT5G56740		77.87	Histone acetyltransferase of the GNAT family 2
BnaC02g10170D	AT5G59050	K18B18.4	77.24	G patch domain protein
BnaC07g16670D	AT5G66550		80.10	Maf-like protein
BnaCnng08340D	unknown			
BnaA06g40000D	unknown			
BnaC07n0282	unknown			
BnaCnn1013	unknown			
BnaC02g22380D	unknown			
BnaA06n0074	unknown			
BnaC07n0181	unknown			
BnaA03n0594	unknown			
J2_24 h vs. J4_24 h\J4_24 h vs. J4_ck	BnaA09 g19610D	AT3G25840	K9I22.6	73.57	Protein kinase superfamily protein
BnaA03g58530D	AT4G21120		90.05	Amino acid transporter 1
J2_24 h vs. J4_24 h\J4_48 h vs. J4_ck	BnaC02g27060D	AT4G02620		87.23	Vacuolar ATPase subunit F protein
J4_24 h vs. J4_ck	BnaC06g34460D	AT1G73650	F25P22.29	88.66	Protein of unknown function (DUF1295)
BnaCnng43580D	AT3G21865		82.39	Peroxin 22
BnaCnng19550D	AT5G57860	MTI20.11	87.37	Ubiquitin-like superfamily protein
BnaA04g17110D	AT5G65940	K14B20.11	80.52	β-hydroxyisobutyryl-CoA hydrolase 1
J4_48 h vs. J4_ck	BnaC08g42380D	AT1G10130	T27I1.16	95.79	Endoplasmic reticulum-type calcium-transporting ATPase 3
BnaC03g69240D	AT1G52490		47.73	F-box/associated interaction domain protein
BnaC09n0143	AT1G62800	F23N19.17	80.31	Aspartate aminotransferase 4
BnaAnng09100D	AT1G75560		85.41	Zinc knuckle (CCHC-type) protein
BnaC08g21130D	AT3G49430	T9C5.30	85.21	Ser/Arg-rich protein 34A
BnaC07g01360D	AT5G03495		30.87	RNA-binding (RRM/RBD/RNP motifs) protein
BnaC04g37090D	AT5G06480		69.11	Immunoglobulin E-set superfamily protein
BnaC08g02660D	unknown			
BnaC09n0005	unknown

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
