# Peer review of "Differential Alternative Splicing Genes and Isoform Regulation Networks of Rapeseed (Brassica napus L.) Infected with Sclerotinia sclerotiorum"

_genes, 2020, doi:10.3390/genes11070784_

Round 1
Reviewer 1 Report
The manuscript analyzed various RNA-seq libraries of mock-inoculated and S. sclerotiorum-inoculated susceptible and tolerant B. napus plants, in order to unravel and discuss the changes in the AS landscape of Brassica napus in response to the fungal pathogen Sclerotinia sclerotiorum.
My overall impression is of a well designed and well discussed work: results, provided in high amount, are clearly presented and properly discussed, without any lack of logical consequence.
I found few minor liguistic issues, which I listed below:
Line 14: "layer" change in "level"
Line 34: "fungal pathogen" after having stated "ascomycete" could be redundant. "fungal" can be deleted without loosing the meaning.
Line 76: "In addition to identifying" change in "In addition, to identify"
Line 171: "were" change in "remained"
Author Response
Dear Reviewers,
Thank you for your kind suggestions and comments. We have made the required corrections. We hope that the revised version of our manuscript will meet with your approval. The main corrections and responses to your comments are listed below in red.
Best regards,
Jiana Li
List of responses:
The manuscript analyzed various RNA-seq libraries of mock-inoculated and S. sclerotiorum-inoculated susceptible and tolerant B. napus plants, in order to unravel and discuss the changes in the AS landscape of Brassica napus in response to the fungal pathogen Sclerotinia sclerotiorum.
My overall impression is of a well designed and well discussed work: results, provided in high amount, are clearly presented and properly discussed, without any lack of logical consequence.
I found few minor liguistic issues, which I listed below:
Line 14: "layer" change in "level"
Response: Thanks for your suggestions! We have changed this word.
Line 34: "fungal pathogen" after having stated "ascomycete" could be redundant. "fungal" can be deleted without loosing the meaning.
Response: Thanks. We have deleted “fungal” as you suggested.
Line 76: "In addition to identifying" change in "In addition, to identify"
Response: Thank you for your suggestion. In order to avoid confusion, we changed it as “In addition to identifying single AS events in a few genes in the past, recent studies have analyzed the massive AS events and explored the genome-wide AS networks.”
Line 171: "were" change in "remained"
Response: Thanks for your suggestions! We have changed it as you suggested.
In the end, thank you for your suggestion again!

Reviewer 2 Report
Peer review
Differential Alternative Splicing Genes and Isoform Regulation Networks of Rapeseed (Brassica napus L.) Infected with Sclerotinia sclerotiorum 

The authors downloaded RNAseq reads from online databases and used them to perform an analysis of splicing variation in differentially resistant and susceptible, and infected and non-infected Brassica napus plants. They describe candidate genes that could explain part of the differential phenotypes observed, and add to evidence that alternative splicing could have a role in disease resistance.
The authors also provide a detailed transcriptional analysis of the candidate genes, and good discussion about the properties of the strongest candidate genes.
I am recommending minor revision, mainly because several of the figures need some improvements to be clear for readers. I also have some minor suggestions to improve the text.
Individual comments:
Introduction
Line 41 “infection of B. napus by S. sclerotiorum follows the ‘gene-for-gene’ model” … “This results in a gradient of resistance phenotypes, termed quantitative disease resistance.”
This explanation is possibly not incorrect but too brief. Simple binary gene for gene interaction does not directly result in quantitative disease resistance. I recommend adjusting to something like: “The outcome of an infection is the product of the interaction of multiple factors in both the host and pathogen, resulting in a gradient of resistance phenotypes, termed quantitative disease resistance”
Line 71: “the AS of many R genes is required for the defense response”
How many? Please be exact, I do not know that many examples.
Figure 1 is OK, but I would like to see the figure modified by adding the treatment and cultivar data above each pie chart, and increasing the font size of the percentage indicators (use lines to indicate for the small slices).
Line 186/187: please be careful. The citation [46] shows that AT5G66890 (homolog of BnaC09g07670D) likely does NOT have a transducing role. I don’t have time to check all your references, but please confirm every assertion that you make from the literature by reading the papers you cite.
Figure 2.- figure legends need a bit more explanation about what is going on here.
Figure 5. Most text is too small, and more obvious labelling will help the reader to understand what is going on.
Discussion is good. Perhaps you could include one or two sentences about what you think should be the next steps to characterize these genes.
Author Response
Dear Reviewers:
Thank you for your kind suggestions and comments. We sincerely appreciate your valuable comments, which helped us improve our manuscript. We have studied your comments carefully and have made the required corrections. We hope that the revised version of our manuscript will meet with your approval. The main corrections and responses to your comments are listed below.
Best regards,
Jiana Li
The authors downloaded RNAseq reads from online databases and used them to perform an analysis of splicing variation in differentially resistant and susceptible, and infected and non-infected Brassica napus plants. They describe candidate genes that could explain part of the differential phenotypes observed, and add to evidence that alternative splicing could have a role in disease resistance.
The authors also provide a detailed transcriptional analysis of the candidate genes, and good discussion about the properties of the strongest candidate genes.
I am recommending minor revision, mainly because several of the figures need some improvements to be clear for readers. I also have some minor suggestions to improve the text.
Individual comments:
Introduction
Line 41 “infection of B. napus by S. sclerotiorum follows the ‘gene-for-gene’ model” … “This results in a gradient of resistance phenotypes, termed quantitative disease resistance.”
This explanation is possibly not incorrect but too brief. Simple binary gene for gene interaction does not directly result in quantitative disease resistance. I recommend adjusting to something like: “The outcome of an infection is the product of the interaction of multiple factors in both the host and pathogen, resulting in a gradient of resistance phenotypes, termed quantitative disease resistance”
Response: Thanks for your suggestion! We really appreciate it that you made this adjustment for us. We have changed the description of quantitative disease resistance as you suggested. “The outcome of SSR is the product of the interaction of multiple factors in both B. napus and S. sclerotiorum, resulting in a gradient of resistance phenotypes, termed quantitative disease resistance.”
Line 71: “the AS of many R genes is required for the defense response”
How many? Please be exact, I do not know that many examples.
Response: Thanks for your advice! We have added the description of AS of R genes. “including TIR-NBS-LRR, Arabidopsis RPS4, Medicago truncatula RCT1, Flax L6 and Tomato Bs4, CC-NBS-LRR and so on, is required for the defense response [27]”
Figure 1 is OK, but I would like to see the figure modified by adding the treatment and cultivar data above each pie chart, and increasing the font size of the percentage indicators (use lines to indicate for the small slices).
Response: Thank you for your comments. We have redrawn Figure 1 to increase the font size with lines indicating small slices and add the description for each pie chart.
Line 186/187: please be careful. The citation [46] shows that AT5G66890 (homolog of BnaC09g07670D) likely does NOT have a transducing role. I don’t have time to check all your references, but please confirm every assertion that you make from the literature by reading the papers you cite.
Response: We apologize for this mistake. We are very sorry about the incorrect function description of AT5G66890 (NRG1C) by confusing it with its paralogs NRG1A and NRG1B. We have rephrased the sentence as “BnaC09g07670D (LRR protein), which is reported likely as a pseudogene with its two paralogs transducing the defense signal downstream of TNL in A. thaliana [47].” We also checked other references. Thanks a lot!
Figure 2.- figure legends need a bit more explanation about what is going on here.
Response: Thanks. We added more explanations to Figure 2. “(b) Soft threshold selection based on the recommended power. (c) Scale free topology check with coefficient of determination (R2) equal to 0.83. (d) Cluster dendrogram of all isoforms by dissimilarity to obtain 85 modules. Each vertical line represents a single isoform.”
Figure 5. Most text is too small, and more obvious labelling will help the reader to understand what is going on.
Response: Thanks for your comments. We have enlarged all the text and labelling in Figure 5.
Discussion is good. Perhaps you could include one or two sentences about what you think should be the next steps to characterize these genes.
Response: Thanks. We added the next research plan as “Continuous advances in molecular biological experiments in exploring the function of different isoforms of one single gene should facilitate our findings of the candidate isoforms in pathogen defense.”
In the end, we are extremely grateful for your helpful suggestions. Thanks!
